# The Essential and the Nonessential Roles of Four Clock Elements in the Circadian Rhythm of *Metarhizium*
*robertsii*

**DOI:** 10.3390/jof8060558

**Published:** 2022-05-25

**Authors:** Han Peng, Yi-Lu Zhang, Sheng-Hua Ying, Ming-Guang Feng

**Affiliations:** Institute of Microbiology, College of Life Sciences, Zhejiang University, Hangzhou 310058, China; 12107005@zju.edu.cn (H.P.); 22007021@zju.edu.cn (Y.-L.Z.); yingsh@zju.edu.cn (S.-H.Y.)

**Keywords:** entomopathogenic fungi, clock proteins, nucleocytoplasmic shuttling, protein–protein interaction, conidiation rhythm

## Abstract

FRQ (frequency protein), FRH (FRQ-interacting RNA helicase), and WC1 and WC2 (white collar proteins) are major clock elements that govern the circadian rhythm in *Neurospora*
*crassa*. However, deletion of *frh* is lethal for the viability of *N. crassa*, making it elusive whether FRH is essential or nonessential for the circadian rhythm. This needs clarification in a fungus where *frh* deletion is not lethal. Here, the nuclear FRH ortholog proved nonessential for the circadian rhythm of *Metarhizium*
*robertsii*. The nucleocytoplasmic shuttling of *M. robertsii* FRQ, WC1, and WC2 orthologs was light-dependent. Yeast two-hybrid assay validated interactions of FRQ with FRH and WC1 instead of FRH with WC1 and WC2 or FRQ with WC2. The circadian rhythm well, shown as conidiation rings of tint and dark in 15 d-old plate cultures grown at 25 °C in a light/dark cycle of 12:12, was abolished in the absence of *frq* or *wc1*, partially disturbed in the absence of *wc2*, but unaffected in the absence of *frh*. These results indicate a requirement of either FRQ or WC1 instead of FRH for the fungal circadian rhythm. Further analyses of *frq* and *frh* mutants revealed the dispensable and the limited roles of FRQ and FRH in the insect-pathogenic lifecycle of *M. robertsii*, respectively.

## 1. Introduction

Aerial conidiation is crucial for the survival and the dispersal of filamentous fungal pathogens in host habitats, and it is indispensable for the lifecycles of imperfect fungi. The model fungus *Neurospora*
*crassa* features rhythmic conidiation that can be readily traced as the rhythm of its circadian clock with carotenoid-pigmented orange conidia [1]. The clock is controlled by a transcription–translation feedback loop composed of an activating positive arm and a repressing negative arm, as well reviewed by [2,3]. The positive arm is governed by two mutually interacting white collar proteins (WC1 and WC2), which form a heterodimeric white collar complex (WCC) responsible for transcriptional activation of FRQ, a key negative-arm frequency protein [4,5,6]. Once induced, FRQ interacts with FRQ-interacting RNA helicase (FRH) and casein kinase 1 (CK1), forming the FRQ-FRH complex (FFC) to interact with WCC [7]. These protein–protein interactions are crucial for the precise timekeeping of the feedback loop in nucleus [8]. In *N. crassa*, a single *frq* gene encodes long and short FRQ isoforms (l-FRQ and s-FRQ) [9]. The ratio of the two isoforms is regulated by thermosensitive splicing at different codons [10,11,12]. Transcription of *frq* is induced by the WCC binding to light-responsive and circadian DNA elements in the *frq* promoter [4,5,13,14], leading to synthesis of FRQ for the formation of functional homodimers via a coiled-coil domain [15,16]. Upon entry into the nucleus, the FRQ homodimers can rapidly repress the transcription of *frq* [8,17,18,19,20] via the FFC-WCC interaction and the FFC-promoted WCC phosphorylation for removal of WCC from the bound DNA elements [19,21,22,23,24,25].

While the feedback loop has been well characterized, it remains obscure in *N. cr**assa* whether FRH is essential or nonessential for the circadian rhythm due to a requirement of *frh* for cell viability. Knockdown mutation of *frh* led to a reduced FRQ level, an elevated *frq* RNA level, and an abolished circadian rhythm, suggesting an essential role for FRH in the circadian rhythm [7]. A small C-terminal region of FRQ was essential for the formation of FFC to ensure the FRQ-WCC interaction and the clock function [26]. The region also proved important for nuclear localization of FRQ [27]. On the other hand, the site-specific mutation *frh^R860H^* was revealed to impair the clock function by affecting an interaction of WCC with FRQ-FRH^R860H^ instead of that of FRQ with FRH^R860H^ [28]. Further mutational analyses uncovered the nonessential role of FRH in the fungal clock despite its role in stabilizing intrinsically disordered FRQ and preventing FRQ degradation [29,30]. The nonessential role of FRH in the circadian rhythm was further revealed by structural analyses of FRH crystal conformations, including the substitutions of R860H disrupting the clock function and V142G alteringthe binding activities of FRQ and WCC to FRH [31]. The previous studies on the mutants of *frh* knockdown and site-mutated alleles unveiled interactions of FRQ with FRH and/or WCC essential for the clock function. However, the features of such mutants bypassing a requirement of *frh* for cell viability may not necessarily reflect the whole role for FRH in the clock. The identification of fungal FRH orthologs that are nonessential for cell viability may offer an opportunity to clarify its role in the circadian rhythm.

Many filamentous fungi possess FRQ and FRH orthologs but are deficient of a distinguishable phenotype like the circadian conidiation rhythm seen in *N. cr**assa*. As an exception, multiple single FRQ domain-containing proteins were found in the proteomic datasets of different *Fusarium*
*oxysporum* strains [32]. Moreover, two FRQ proteins (Frq1 and Frq2) different in molecular size coexist in *Beauveria*
*bassiana* (Cordycepitaceae, Hypocreales). Opposite nuclear dynamics (rhythms) of Frq1 and Frq2 were proven to persistently activate the key activator genes of the central developmental pathway in a circadian day to support the mode of nonrhythmic conidiation, which ensures rapid maximization of conidial production in plate cultures [33] and on insect cadaver surfaces [34]. The opposite dynamics of Frq1 and Frq2 in the nucleus were abolished by a knockout mutation of the *frh* ortholog, leading to restoration of aerial conidiation to a wild-type level that was sharply reduced in the absence of *frq1* or *frq2* [33]. These studies suggest diversity and complexity of circadian systems in different lineages of filamentous fungi aside from the model studied intensively in *N. crassa*. However, the nonrhythmic conidiation of *B. bassiana* orchestrated by Frq1 and Frq2 in the presence or absence of *frh* is helpless to clarify the essential or the nonessential role of FRH in the timekeeping of the feedback loop. The clarification is reliant upon whether the circadian rhythm is complete or abolished in the absence of *frh*. To explore such evidence, we characterized four clock elements in *Metarhizium*
*robertsii* (Clavicipitaceae), another hypocrealean insect pathogen that has FRQ, FRH, WC1, and WC2 orthologs and that displays a typical circadian rhythm as conidiation rings of tint and dark in plate cultures in a light/dark (L:D) cycle of 12:12. The fungal circadian rhythm is well shown in a film visualization system. Previously, mutually interacting WC1 and WC2 orthologs were proven to interact with both Phr1 and Phr2, two photolyases required for photorepair of UV-induced DNA lesions and photoreactivation of UV-inactivated conidia, and to act as core regulators of UV resistance in *M. roberstii* [35]. In this study, we analyzed subcellular localization and protein–protein interactions of the four clock elements, the phenotypes of single-gene knockout mutants, and the effects of each knockout on the transcript levels of the three other genes in different L:D cycles. As presented below, the fungal circadian rhythm was fully abolished in the absence of *frq* or *wc1*, partially disturbed in the absence of *wc2*, but not affected at all in the absence of *frh*. These phenotypes and the confirmed interactions of FRQ with FRH and WC1 instead of either FRH with WC1 and WC2 or FRQ with WC2, uncover a circadian clock similar to the model in *N. cr**assa* but suggest a nonessential role for FRH in the circadian rhythm of *M. roberstii*.

## 2. Materials and Methods

### 2.1. Microbial Strains and Culture Conditions

The wild-type strain *M. robertsii*ARSEF 2575 (designated WT) was incubated on 1/4 SDAY (one-fourth nutrition strength of Sabouraud dextrose agar: 1% glucose, 0.25% peptone, and 1.5% agar plus 0.25% yeast extract) or PDA (potato dextrose agar) at 25 °C and used as a recipient of targeted gene manipulation. Fungal stress assays were performed on CDA (Czapek-Dox agar: 3% sucrose, 0.3% NaNO_3_, 0.1% K_2_HPO_4_, 0.05% KCl, 0.05% MgSO_4_, and 0.001% FeSO_4_ plus 1.5% agar) at 25 °C. *Escherichia coli* DH5α and TOP10 were cultured in Luria-Bertani medium at 37 °C for vector propagation. *Agrobacterium tumefaciens* AGL1 was cultured at 28 °C in a broth composed of 0.5% sucrose, 1% peptone, 0.1% yeast extract, and 0.05% MgSO_4_, and it was used for fungal transformation [36]. *Saccharomyces cerevisiae* Y187 and Y2HGold were used for yeast two-hybrid (Y2H) assays.

### 2.2. Recognition and Bioinformatic Analysis of Fungal FRQ and FRH Orthologs

The amino acid sequences of *N. crassa* FRQ (XP_011395125) and FRH (XP_956298) were used as queries to search through the NCBI databases of *M. robertsii* [37] and other 25 entomopathogenic and nonentomopathogenic ascomycetes by BLASTp analysis (http://blast.ncbi.nlm.nih.gov/blast.cgi, accessed on 5 May 2022). The identified homologs were subjected to phylogenetic analysis with a maximum likelihood method in the online program MEGA7 (http://www.megasoftware.net, accessed on 5 May 2022). The FRQ and the FRH homologs found in *M. robertsii* and *B. bassiana* were structurally compared with the two queries, respectively, by conserved domain analysis at http://smart. embl-heidelberg.de/ (accessed on 5 May 2022), followed by the prediction of a nuclear localization signal (NLS) motif from each protein sequence with maximal probability at https://www.novopro.cn/tools/nls-signal-prediction(accessed on 5 May 2022).

### 2.3. Generation of frq and frh Mutants

The previous strategy for generation of *wc1* and *wc2* mutants [35] was adopted to delete *frq* or *frh* in the WT strain. Briefly, the 5′ and 3′ fragments of *frq* or *frh* were amplified from the genomic DNA of the WT strain and then inserted into appropriate enzyme sites of linearized p0380-bar, yielding p0380- 5′*x*-bar-3′*x* (*x* = *frq* or *frh*) for targeted gene deletion. The full-length coding sequence of each target gene with flank regions was amplified from the WT DNA and ligated into p0380-sur-gatewayto replace the gateway fragment under the action of Gateway BP Clonase II Enzyme Mix (Invitrogen, Shanghai, China), resulting in p0380-sur-*x* for targeted gene complementation. Each target gene was deleted from the WT strain by homologous recombination of the *bar*-separated 5′ and 3′ coding/flanking fragments and then complemented into an identified deletion mutant by ectopic integration of the cassette comprising its full-length coding/flanking sequence and the *sur* marker via *Agrobacterium*-mediated transformation [36]. Putative deletion and complementation mutants were screened by the *bar* resistance to phosphinothricin (200 μg/mL) and the *sur* resistance to chlorimuron ethyl (10 μg/mL), respectively, followed by PCR identification of expected recombination events and verification of targeted gene deletion or complementation by real-time quantitative PCR (qPCR) analysis. Listed in Appendix A are paired primers used for the manipulation and the detection of each target gene. The deletion mutants (DM) Δ*frq* and Δ*frh* and the complementation mutants (CM) Δ*frq::frq* and Δ*frh::frh* were evaluated together with the parental WT in experiments of three independent replicates to meet a requirement for one-way analysis of variance.

### 2.4. Transcriptional Profiling

The present *frq* and *frh* mutants, the previous *wc1 and*
*wc2* mutants [35], and the WT strain were grown on cellophane-overlaid PDA plates by spreading 100 μL of a 10^7^ conidia/mL suspension per plate and incubated for 3 d at 25 °C in the L:D cycles of 0:24, 12:12 and 24:0 to assess the impact of each knockout mutation on the expression levels of the deleted gene and the three other genes. For transcriptional profiling of *frq* and *frh* in the WT strain, the PDA cultures were incubated for 7 d at the optimal regime of 25 °C and L:D 12:12. Total RNAs were separately extracted from the WT cultures daily during the period of 7-d incubation or from the 3 d-old cultures of each strain grown in each L:D cycle with an RNAiso Plus Kit (TaKaRa, Dalian, China) and reversely transcribed into cDNAs with a Prime ScriptH^RT^ reagent kit (TaKaRa). Three cDNA samples (standardized by dilution) derived from the cultures were used as templates to quantify: (1) daily transcript levels of *frq* or *frh* in the WT strain during the 7-d incubation with respect to a standard on day 2, and (2) transcript levels of *frq*, *frh*, *wc1*, or *wc2* in the 3 d-old cultures of each DM or CM with respect to the WT standard in each L:D cycle. The qPCR analysis with paired primers (Appendix A) was performed with MonAmp^M^ SYBR^®^Green qPCR Mix (Monad Biotech Co., Suzhou, China). The fungal 18S rRNA was used as an internal standard. The relative transcript level of each gene was computed with a threshold-cycle (2^−ΔΔCt^) method.

### 2.5. Subcellular Localization of FRQ and FRH in M. robertsii

Recombinant strains expressing the green fluorescence protein GFP-tagged FRQ and FRH fusions in the WT strain were constructed as described previously for subcellular localization of WC1 and WC2 [35]. Briefly, the coding sequence of *frq* or *frh* was amplified from the WT cDNA and ligated to the N-terminus of *gfp* in linearized p0380-C-sur, where C denotes the cassette 5′-EcoRI-XmaI-BamHI-PstI-HindIII-3′ under the control of the endogeneous promoter P*tef1*. Each coding sequence was ligated to the enzyme site of *Xma*I or of *Pst*I. The WT strain was transformed withthe plasmids p0380- frq-gfp-sur and p0380-frq-gfp-sur as aforementioned. Further, p0380-frq-gfp-sur was ectopically integrated into the Δ*frh* mutant generated as aforementioned and the Δ*wc1* and Δ*wc2* mutants constructed previously [35], respectively. Putative transformants were screened by the *sur* resistance to chlorimuron ethyl (10 μg/mL). For each transformation, a transgenic strain displaying a strong green fluorescence signal was chosen to incubate on 1/4 SDAY for conidiation at the optimal regime of 25 °C and L:D 12:12. Conidia collected from the culture were suspended in SDBY (agar-free SDAY) and incubated for 3 d on a shaking bed (150 rpm) at 25 °C in the L:D cycles of 0:24 (full dark), 12:12 and 24:0 (full light) (for FRQ-GFP or FRH-GFP in WT) and in the L:D cycle of 12:12 alone (FRQ-GFP in Δ*frh*, Δ*wc1* and Δ*wc2*), respectively. Hyphal samples collected from each culture were stained with 4.16 mM of the nuclear dye DAPI (4′,6′-diamidine-2′- phenylindoledihydrochloride; Sigma-Aldrich, Shanghai, China) and visualized with laser scanning confocal microscopy (LSCM) at the excitation/emission wavelengths of 358/460 and 488/507 nm respectively to determine subcellular localization of FRQ-GFP and FRH-GFP in WT and of FRQ-GFP expressed in the knockout mutants. The software ImageJ (https://imagej.nih.gov/ij/, accessed on 5 May 2022) was used to measure green fluorescence intensities from a fixed circular area moving in the cytoplasm and the nuclei of 15 hyphal cells from each of the cultures with FRQ-GFP expressed in the WT strain grown in the three L:D cycles or expressed in the absence of *frh*, *wc1*, or *wc2* at L:D 12:12. The ratios of nuclear versus cytoplasmic green fluorescence intensities (N/C-GFI) were computed as relative accumulation levels of expressed FRQ-GFP in the nuclei under different conditions.

### 2.6. Y2H Assays

Protein–protein interactions of FRQ with FRH and of FRQ or FRH with WC1 or WC2 were tested following Matchmaker GAL4 Two-Hybrid System 3 & Libraries User Manual (http://www.clontech.com, accessed on 5 May 2022). Briefly, the coding sequences of *frq* (MAA_04673), *frh* (MAA_08584), *wc1* (MAA_04453), and *wc2* (MAA_07440) were amplified from the WT cDNA and inserted into the prey vector pGADT7 (AD) or the bait vector pGBKT7 (BD), followed by sequencing for verification. Each verified construct was transformed into the *S. cerevisiae* Y187 and Y2HGold strains, respectively, and incubated for 24 h of pairwise yeast mating at 30 °C on YPD (1% yeast extract, 2% peptone, 2% glucose plus 0.04% adenine hemisulfate salt). The diploids (AD-FRQ-BD-FRH, AD-WC1-BD-FRH, AD-WC2-BD-FRH, AD-WC1-BD-FRQ, and AD-WC2-BD-FRQ) were screened in parallel with positive control (AD-LargeT-BD-P53) and negative controls (AD-BD and the constructs expressing AD-BD-FRH, AD-FRQ-BD, AD-BD-FRQ, AD-WC1-BD, and AD-WC2-BD) on the double-dropout (SD/-Leu/-Trp/X-α-Gal/AbA) and quadruple-dropout (SD/-Leu/-Trp/-Ade/-His/X-α- Gal/AbA) plates of a synthetically defined medium (SD), respectively. The yeast colonies bearing each construct were initiated by spotting 1 μL aliquots of 10^5^, 10^6^, or 10^7^ cells/mL suspensions, followed by a 3-d incubation at 30 °C for image records.

### 2.7. Observation of Conidiation Rhythm

The PDA cultures of the DM and the WT strains were initiated by spotting 1 μL of 10^6^ conidia/mL suspension at the center of each plate and then incubated for up to 15 d at optimal 25 °C in the L:D cycle of 12:12 (positive control) versus 0:24 and 24:0 (negative control). At the end of incubation, the plate cultures were inversely placed on the translucent glass board (illuminated below) of a film visualization system (device), and they were individually photographed to show changes in the circadian rhythm featured by conidiation rings of tint and dark around the center.

### 2.8. Assays for Stress Response, Conidial Yield, UVB Resistance, and Virulence

The DM strains of *frq* and *frh* and their control (WT and CM) strains were grown by spotting 1 μL aliquots of a 10^6^ conidia/mL suspension on CDA plates filled with H_2_O_2_ (4.41 mM) or menadione (30 μM) for oxidative stress, NaCl (0.4 M) or sorbitol (1 M) for osmotic stress, and Congo red (2.87 mM), calcofluor white (40 μg/mL) or sodium dodecyl sulfate (30 μM) for cell wall perturbing stress, respectively, or not filled with any chemical stressor. After a 7-d incubation at the optimal regime of 25 °C and L:D 12:12, the diameter of each colony was assessed as a growth index with two measurements taken perpendicularly to each other across the center.

The cultures of each strain for assessment of conidial yield were initiated by spreading 100 μL of a 10^7^ conidia/mL suspension per 1/4 SDAY plate (9 cm diameter) and then incubating for 15 d at the optimal regime. Three samples were taken from each plate culture using a cork borer (5 mm diameter) and individually placed in 1 mL aliquots of aqueous 0.05% Tween 80, followed by 10 min supersonic vibration for the release of conidia. The concentration of conidia in the resultant suspension was determined with a hemocytometer and converted to the number of conidia per square centimeter of plate culture. In addition, conidial resistance to UVB irradiation (weighted 312 nm wavelength) was assessed as previously described [35].

The fifth-instar larvae of greater wax moth (*Galleria mellonella*) were assayed for the virulence of each strain in two infection modes. Briefly, three groups of ~35 larvae per strain were immersed for 10 s in 40 mL aliquots of a 10^7^ conidia/mL suspension for normal cuticle infection. Alternatively, a microinjector was used to inject 5 μL of a 10^5^ conidia/mL suspension into the hemocoel of each larva in each group for cuticle-bypassing infection. After inoculation, all groups of larvae were maintained at 25 °C for survival or mortality records every 12 or 24 h until no more change in mortality. The time–mortality trend in each group of larvae infected by topical application or intrahemocoel injection was analyzed to estimate median lethal time (LT_50_) as an index of virulence.

## 3. Results

### 3.1. Structural and Phylogenetic Features of Fungal FRQ and FRH Orthologs

BLASTp analyses using the queries of the *N. cr**assa* FRQ and FRH sequences resulted in identification of FRQ and FRH orthologs in most or all surveyed fungal genomes, respectively. Exceptionally, two, three, and multiple (>10) FRQ homologs were found in the genomes of *B. bassiana* ARSEF 2860 (964 and 583 aa, previously characterized as Frq1 and Frq2 [33]), *Fusarium*
*gramin**earum* PH-1 (993, 314 and 48 aa) and *F. oxysporum* (480–992 aa). Almost all of these FRQ homologs contain a single FRQ domain covering most of each protein sequence. The *M. robertsii* FRQ (XP_007820960,1005 aa) are similar in molecular size and FRQ domain to the query or *B. bassiana* Frq1 but distinctive from *B. bassiana* Frq2, which features two FRQ domains adjacent to each other and much smaller molecule (Figure 1A). The *M. robertsii* FRH (XP_007824773, 1098 aa) resembles the *B. bassiana* and *N. cr**assa* orthologs in molecular size and structure, sharing the major domains DEXDc (DEAD-like helicases superfamily), HELICc (helicase superfamily C-terminal domain), rRNA_proc-arch required for proper 5.8S rRNA processing [38], and DSHCT (DOB1/SK12/helY-like DEAD box helicases). An NLS motif was predicted in each of the FRH sequences at higher maximum probabilities (0.851–0.897) than in each of the FRQ sequences (0.351–0.819). In phylogeny, either FRQ (Figure 1B) or FRH orthologs (Figure 1C) were clustered to clades and subclades associated with fungal lineages, leading to much higher sequence identities of *M. robertsii* FRQ and FRH to hypocrealean counterparts (59–99% and 82–100%) than to non-hypocrealean orthologs (34–52% and 61–71%).

As an exception, three single FRQ domain-containing proteins in *F*. *gramin**earum* PH-1 fell into three subclades, and the smallest homolog (48 aa) was clustered together with the *B. bassiana* Frq2. Multiple FRQ-like proteins with variable molecular size in *F. oxysporum* were not included in the phylogenetic analysis.

### 3.2. Transcriptional Profiles and the Subcellular Localization of FRQ and FRH in M. robertsii

Both *frq* and *frh* were constitutively expressed in the WT strain during a 7-d incubation on PDA at the optimal regime, and markedly upregulated at a transcriptional level in comparison to a standard on day 2 (Figure 2A). The time-course expression level of *frq* fluctuated at a greater magnitude than that of *frh*, which was upregulated by 4- to 9-fold relative to the standard.

The FRH-GFP fusion protein expressed in the WT strain accumulated exclusively in the nuclei of hyphae stained with the nuclear dye DAPI irrespective of being cultured in the L:D cycle of 0:24, 12:12 or 24:0 (Figure 2B). These observations coincide well with an NLS motif predicted from the FRQ sequence at the high probability of 0.85, and they indicate a light-independent localization of FRH in the nucleus. The GFP-tagged FRQ fusion protein expressed in the WT strain accumulated in the nuclei more than in the cytoplasm of hyphal cells grown in the three L:D cycles (Figure 2C). The ratios of nuclear versus cytoplasmic green fluorescence intensities (N/C-GFI) of FRQ-GFP in the hyphal cells (Figure 2D) were averaged as 10.9 (±3.4) at L:D 12:12, and this mean (±SD) ratio was significantly greater (*p* < 0.01 in Tukey’s HSD test) than that at L:D 0:24 (7.7 ± 2.6) or 24:0 (4.8 ± 1.4). These ratios implicate a main localization of FRQ in nuclei and also a dependence of its nucleocytoplasmic shuttling on appropriate light exposure as was shown for the shuttling of WC1 or WC2 in the previous study [36].

Next, FRQ-GFP was expressed in different knockout mutants to assess the impacts of three other clock proteins on the nucleocytoplasmic shuttling of FRQ at L:D 12:12. The fusion protein accumulated in both the nuclei and the cytoplasm of the hyphae from the Δ*f**rh*, Δ*wc1,* or Δ*wc2* culture (Figure 2E). The mean N/C-GFI ratios were 3.2 (±1.1), 2.6 (±0.7) and 2.5 (±0.6) for FRQ-GFP expressed in the Δ*frh*, Δ*wc1*, and Δ*wc2* hyphae, respectively, (Figure 2F) but showed an insignificant variability among the genetic backgrounds (*F*_2,42_ = 3.13, *p* = 0.054). These mean ratios were sharply reduced in comparison to the estimate of the fusion protein expressed in the WT strain at L:D 12:12. The N/C-GFI ratios reduced in the absence of *frh*, *wc1,* or *wc2* strongly suggest that FRH, WC1, and WC2 may exert similar effects on the nucleocytoplasmic shuttling of FRQ at L:D 12:12.

### 3.3. Interactions of FRQ with FRH and WC1

A positive interaction between WC1 and WC2 for the formation of WCC was confirmed previously in *M. robertsii* [35]. In the present Y2H assay, an interaction of FRQ with FRH for the formation of FFC was validated by the constructed yeast diploid AD-FRQ-BD-FRH that grew as well on the quadruple-dropout plate as the positive control AD-LargeT-BD-P53 (Figure 3A). In contrast, FRH was found to not interact with either WC1 or WC2. The constructed diploids also showed an interaction of FRQ with WC1 but not with WC2 (Figure 3B). The validated interactions suggest an essential role of FRQ instead of FRH in the feedback loop characterized in *N. cr**assa* [2,3].

### 3.4. Circadian Rhythm Is Dependent on FRQ but Independent of FRH

The circadian rhythm shown as conidiation rings of tint and dark in the film visualization system was present in the 15 d-old PDA cultures of the WT strain grown at 25 °C and L:D 12:12 but not at L:D 0:24 or 24:0 (Figure 4A). The rhythm at L:D 12:12 was completely abolished in the Δ*frq* and Δ*wc1* cultures, partially disrupted in the Δ*wc**2* cultures, but not at all affected in the Δ*frh* cultures. These observations indicated a dependence of the circadian rhythm on FRQ and FRQ-interacting WC1 rather than on FRH in *M. robert**sii*. The lesser effect of *wc2* deletion on the rhythm could be due to no impact of deleted *wc2* on the existence of WC1 and FRQ essential for the FFC-WCC interaction.

Next, we analyzed transcript levels of clock protein-coding genes in the cDNA samples of each DM and control strains derived from the 3 d-old PDA cultures incubated at 25 °C in the three L:D cycles. In general, the deletion of one clock gene resulted in differential expressions of the three others in the L:D cycles. In the Δ*frh* mutant, expression of *frq* was nearly abolished in the dark and restored to half and normal levels of the WT strain in the respective L:D cycles of 12:12 and 24:0 (Figure 4B); *wc1* was significantly downregulated in the dark but upregulated with increasing light exposure while *wc2* was upregulated under light but unaffected at L:D 0:24 or 12:12. The deletion of *frq* resulted in significant up-regulation of *frh* or *wc1* in the dark and significant down-regulation of *wc1* and *wc2* under light and of *wc2* at L:D 12:12 (Figure 4C). As a result of *wc1* deletion, *frh* and *wc2* were significantly suppressed in the dark but upregulated in the L:D cycles of 0:12 and 12:12, accompanied by a reduced expression of *frq* at L:D 12:12 (Figure 4D). The *wc2* deletion led to significant repression of only *frq* under light but upregulated expression of *frq* in the dark and of *wc1* at L:D 12:12 (Figure 4E).

All transcript changes in the DM strains were restored to the WT levels by targeted gene complementation. These data demonstrated light-dependent impacts of each deleted gene on the expression levels of three others. Notably, the presence of full or partial conidiation rhythm in the Δ*frh* or Δ*wc2* cultures correlated with half of the WT’s *frq* expression level at L:D 12:12. The abolished circadian rhythm in the Δ*wc1* cultures with *frq* expressed at half of the WT level at L:D 12:12 reinforced the importance of FRQ-WC1 interaction for the fungal clock function.

### 3.5. Dispensable and Limited Roles of FRQ and FRH in the Fungal Lifecycle

The roles of *wc1* and *wc2* in the asexual cycle and the virulence of *M. rober**tsii* have been shown in the previous study [35]. In the present study, the deletion of *frq* resulted in little impact on radial growth on CDA alone or supplemented with different types of chemical stressors (Figure 5A), conidial yields in the 15 d-old cultures grown on 1/4 SDAY in different L:D cycles (Figure 5B), and conidial resistance to UVB irradiation (Figure 5C). Exceptionally, the Δ*frq* mutant showed moderate but significant reductions in cell tolerance to oxidative stress induced by menadione (30 μM) and a conidiation level at L:D 12:12. In contrast, the Δ*frh* mutant displayed more, but limited, phenotypic changes. Its growth was defective on CDA, and suppressed more significantly under the stresses of Congo red (2.87 mM) and calcofluor white (40 μg/mL) than of H_2_O_2_ (4.41 mM), sorbitol (1 M), and SDS (30 μM). Its conidial yield was reduced by 18% in the dark and 40% at L:D 12:12 in comparison to the corresponding WT yields, accompanied by 35% decrease in conidial UVB resistance. In the standardized bioassays, however, the fungal virulence indicated by the LT_50_ estimates was affected insignificantly or marginally for the Δ*frq* and Δ*frh* mutants against the model insect via normal cuticle infection or cuticle- bypassing infection (Figure 5D). These data demonstrated the limited role of FRH but the nearly dispensable role of FRQ in the asexual cycle in vitro and the infection cycle of *M. robertsii*.

## 4. Discussion

Our experimental data demonstrate a light-independent localization of FRH in nucleus and a light-dependent nucleocytoplasmic shuttling of FRQ in *M. robertsii* as shown previously in *B. bassiana* [33]. In *M. robertsii*, WC1 and WC2 also shuttled between nucleus and cytoplasm in a light-dependent fashion, but they accumulated more consistently in the nucleus than in the cytoplasm [35]. The fungal circadian rhythm well shown in the film visualization system was present in the WT and the Δ*frh* cultures but abolished in the Δ*frq* and Δ*wc1* cultures at L:D 12:12, but never appeared in the cultures grown under full light or in full darkness. The photoperiod of L:D 12:12 led to a maximum of nuclear FRQ accumulation in the WT strain and an approximate half expression of *frq* in the absence of *frh*, *wc1,* or *wc2*. In the Δ*frh* mutant, the half expression of *frq* had no effect on not only the circadian rhythm but also the FRQ-WC1 interaction that led to a considerably high accumulation level of FRQ in the nucleus at L:D 12:12. These results uncover the essential roles of both FRQ and FRQ-interacting WC1 in the circadian rhythm of *M. robertsii*. The role of FRH is seemingly limited to its interaction with FRQ alone for the formation of FFC as revealed in the present Y2H assay. FRH also functions in sustaining the opposite nuclear dynamics of two FRQs in *B. bassiana* [33], stabilizing intrinsically disordered FRQ, and preventing FRQ from degradation in *N. cr**assa* [29,30]. The role of WC2 to interact with neither FRQ nor FRH is limited to its interaction with WC1 for the formation of WCC in *M. robertsii* [35], thereby leading to a disturbed, but not abolished, circadian rhythm when the interaction was inhibited in the absence of *wc2*. Importantly, both WC1 and WC2 have no DNA_photolyase domain, but they can mediate photorepair of DNA lesions and photoreactivation of UV-impaired conidia by interactions of either with two photolyases in *M. robertsii* [35], leading to the hypothesis that a WCC-cored pathway may exist in the filamentous fungal response and resistance to solar UV irradiation [39]. Despite no direct link to the circadian rhythm, FRH and WC2 act as components of FFC and WCC, respectively, and hence are important players in the clock’s feedback loop. This is because FFC can interact with WCC and facilitate the phosphorylation of WCC for the removal of WCC from the bound DNA elements of *frq* [19,21,22,23,24,25].

Aside from an essential role in the circadian rhythm, FRQ was dispensable for almost all examined phenotypes in *M. robertsii*. This is very different from the roles of Frq1 and Frq2 in mediating conidiation capacity, insect-pathogenic lifecycle, and calcofluor-specific signal transduction through the mitogen-activated protein kinase (MAPK) Slt2 cascade in *B. bassiana* [33,34]. In particular, the previous Δ*frq1* and Δ*frq2* mutants showed hypersensitivity and high resistance to cell wall stress induced by calcofluor white and hypo- and hyper-phosphorylation of the MAPK Slt2 required for cell wall integrity [34,40], respectively. The dispensable role of FRQ in the virulence of *M. robertsii* is also different from the regulatory role of orthologous FRQ (BcFRQ1) in the virulence of *Botrytis cinerea* against *Arabidopsis thaliana* [41]. In contrast, the present Δ*frh* mutant displayed moderate defects in normal colony growth, conidiation at L:D 0:24 or 12:12, cellular tolerance to stresses induced by H_2_O_2_, three cell wall perturbing agents, and conidial resistance to UVB irradiation. These phenotypic changes indicate a limited role of FRH in the asexual cycle in vitro of *M. robertsii*, but they are different from the insignificant effect of *frh* deletion on the main phenotypes of *B. bassiana* [33]. The present and the previous studies unveil, importantly, the differential roles of FRQ and FRH in the lifecycles of these two hypocrealean insect pathogens. This is likely due to a 130-MY difference in the evolutionary history of insect-pathogenic lifestyle between *Beauveria* and *Metarhizium* [42,43].

## 5. Conclusions

The circadian rhythm relies directly upon a feasibility of FRQ-WC1 interaction in *M. robertsii*. This is unveiled by the abolished rhythm concurring with the interaction inhibited in the absence of *frq* or *wc1* and the perfect rhythm concurring with the interaction not affected in the absence of *frh*. In contrast with the essential roles of FRQ and WC1, FRH is nonessential for the circadian rhythm although FRH and WC2 are players in the whole feedback loop. Our findings support the previous conclusion that was drawn through analyses of those FRH site-specific mutants in *N. cr**a**ssa* [28,29,30,31].

## Figures and Tables

**Figure 1 jof-08-00558-f001:**
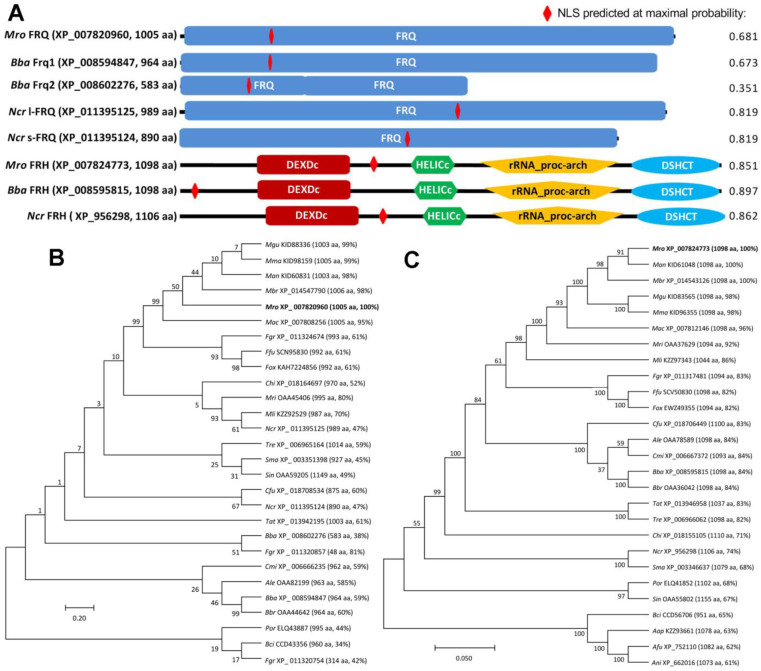
Sequence features and phylogenetic linkages of fungal FRQ and FRH homologs. (**A**) Comparison of conserved domains and NLS motifs predicted from the FRQ and FRH sequences of *M. robertsii* (*Mro*), *B. bassiana* (*B**ba*) and *N. crassa*(*Ncr*). A maximal probability associated with the NLS motif predicted is shown at the end of each sequence. (**B**,**C**) Phylogenetic linkages of FRQ and FRH homologs in ascomycetous insect pathogens (*Ale, Akanthomyces*
*lecanii*; *Aap*, *Ascosphaera*
*apis*; *Bbr, B**eauveria*
*brongniartii*; *Cfu, Cordyceps*
*fumosorosea*; *Cmi*, *C. militaris*; *Mac*, *Metarhizium*
*acridum*; *Man*, *M. anisopliae*; *Mbr*, *M. brunneum*; *Mgu*, *M. guizhouense*; *Mma*, *M. majus*; *Mri*, *M. rileyi*; *Mli*, *Moelleriella*
*libera*; *Sin*, *Sporothrix*
*insectorum*) and non-entomopathogenic fungi (*Afu, Aspergillus*
*fumigatus*; *Ani, A. nidulans*; *Bci, Botrytis cinerea*; *Chi, Colletotrichum*
*higginsianum*; *Ffu*,*Fusarium*
*fujikuroi*; *Fgr*, *F. graminearum*; *Fox*, *F. oxysporum*; *Por*, *Pyricularia*
*oryzae*; *Sma*, *Sordaria*
*macrospora*; *Tat*, *Trichoderma*
*atroviride*; *Tre*, *T*
*.reesei*), respectively. The bootstrap values of 1000 replications are given at nodes. Scale bar: branch length proportional to genetic distance assessed with the maximum likelihood method in MEGA7.

**Figure 2 jof-08-00558-f002:**
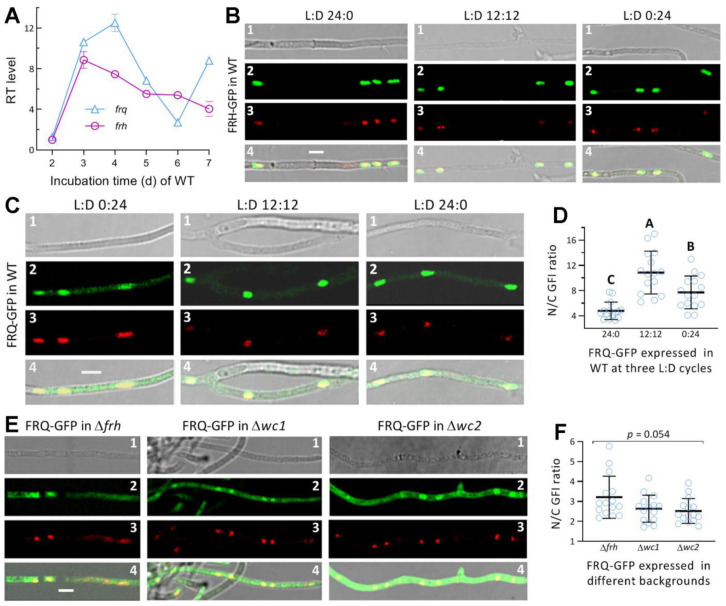
Transcriptional profiles and subcellular localization of FRQ and FRH in *M. robertsii*. (**A**) Relative transcript (RT) levels of *frq* and *frh* in the WT strain during a 7-d incubation on PDA at the optimal regime of 25 °C and L:D 12:12, normalized with the expression levels detected on day 2. (**B**,**C**) LSCM images (scale bars: 5 μm) for subcellular localization of GFP-tagged FRQ and FRH fusion proteins expressed in the WT strain. Cell samples were taken from the 3 d-old SDBY cultures grown at 25 °C in the L:D cycles of 0:24, 12:12, and 24:0 and stained with the nuclear dye DAPI (shown in red). (**D**) Nuclear versus cytoplasmic green fluorescence intensity (N/C-GFI) ratios of FRQ-GFP in the hyphal cells. Different uppercase letters denote significant differences (*p* < 0.01 in Tukey’s HSD test). (**E**,**F**) LSCM images (scale bars: 5 μm) and N/C-GFI ratios for subcellular localization of FRQ-GFP expressed in the 3 d-old SDBY cultures of the Δ*frh*, Δ*wc1* and Δ*wc2* mutants at 25 °C and L:D 12:12, respectively. Images 1–4 in each panel are bright, expressed (green), stained (red nuclei) and merged (yellow nuclei) views of the same field, respectively. Error bars: standard deviations (SDs) of the means from three cDNA samples (**A**) analyzed via qPCR or 15 cells in the examined hyphae (**D**,**F**).

**Figure 3 jof-08-00558-f003:**
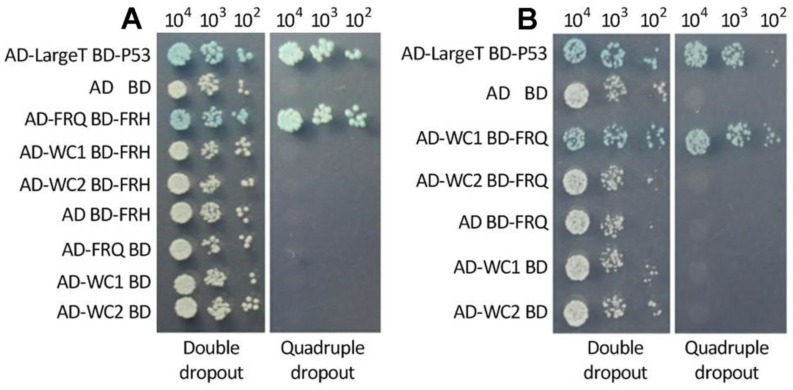
Y2H assay for protein–protein interactions. A positive interaction is indicated by the same growth capability of the constructed diploid AD-FRQ-BD-FRH (**A**) or AD-WC1-BD-FRQ (**B**) as the positive control (AD-LargeT-BD-P53) on the quadruple-dropout medium (SD/-Leu/-Trp/- Ade/-His/X-α-Gal/AbA). The colonies of each cell type were initiated with 1 μLaliquots of 10^5^, 10^6^, and 10^7^ cells/mL suspensions and incubated for 3 d at 30 °C.

**Figure 4 jof-08-00558-f004:**
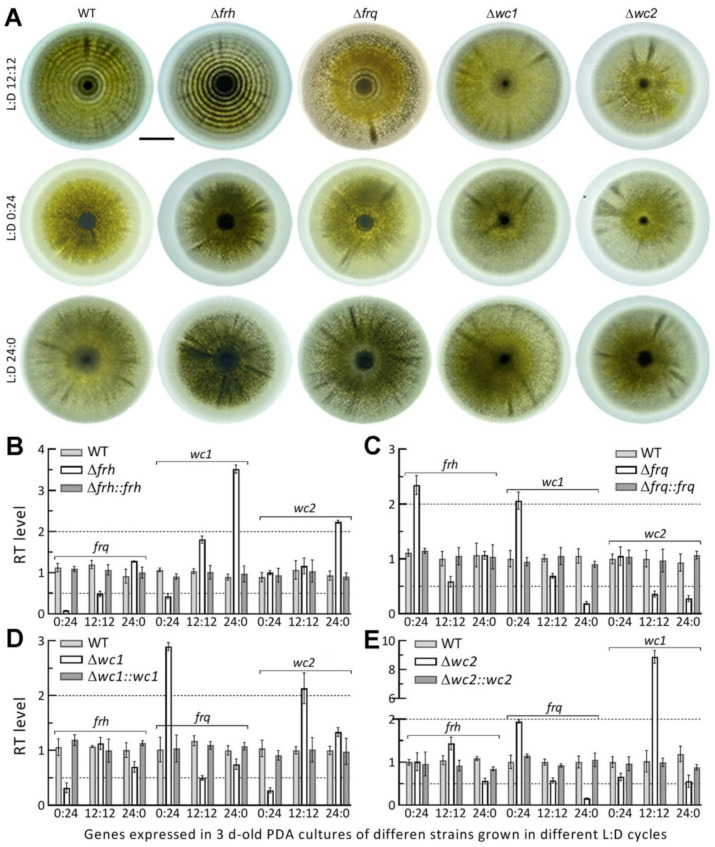
Impacts of knockout mutations on the circadian rhythm of *M. robertsii*. (**A**) Images (scale bar: 10 mm) for thecircadian rhythm shown (in a film visualization system) in the 15 d-old PDA cultures of the WT, Δ*frh*, Δ*frq*, Δ*wc1,* and Δ*wc2* strains incubated at 25 °C in the indicated L:D cycles. (**B**–**E**) Effects of *frh*, *frq*, *wc1,* or *wc2* deletion on relative transcript (RT) levels of the three other genes in the 3 d-old PDA cultures in the L:D cycles. Upper and lower dashed lines denote one-fold up- and down-regulation versus the WT strain. Error bars: SDs from three independent cDNA samples.

**Figure 5 jof-08-00558-f005:**
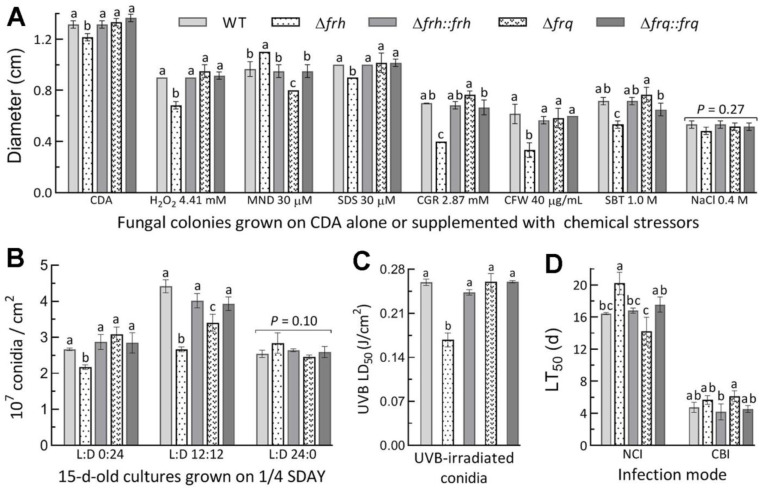
Effects of *frq and frh* knockout mutations on phenotypes associated with the lifecycle of *M. robertsii*. (**A**) Diameters of fungal colonies incubated at 25 °C on CDA plates alone or supplemented with indicated concentrations of chemical stressors (MND, menadione; SDS, sodium dodecyl sulfate; CGR, Congo red; CFW, calcofluor white; SBT, sorbitol). Each colony was initiated with 10^3^ conidia. (**B**) Conidial yields measured from the 15 d-old cultures at 25 °C in the indicated L:D cycles. (**C**) The estimates of median lethal dose (LD_50_) for conidial resistance to UVB irradiation. (**D**) The LT_50_ estimates for the knockout and control strains against *G*. *mellonella* larvae inoculated by topical application of a 10^7^ conidia/mL for normal cuticle infection (NCI) or intrahemocoel injection of ~500 conidia per larva for cuticle-bypassing infection (CBI). Different lowercase letters in each bar group denote significant differences (*p* < 0.05 in Tukey’s HSD tests). Error bars: SDs of the means from three independent replicates.

## Data Availability

All experimental data are included in this paper and Appendix A.

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
