# Peer review of "The Essential and the Nonessential Roles of Four Clock Elements in the Circadian Rhythm of *Metarhizium"

_jof, 2022, doi:10.3390/jof8060558_

Round 1
Reviewer 1 Report
In this article the authors evaluated four elements of the circadian cycle in Metarhizium robertsii described in other fungi such as Neurospora. The introduction, methodology, results and discussion, are adequate for the research. It is the first time in M. robertsii that the role of FRQ (frequency protein) and FRH (FRQ-interacting RNA helicase) proteins has been analyzed. Conclusions can be improved.
Author Response
Please see attached a file.

Reviewer 2 Report
In this manuscript, the authors characterize the phenotypes of the single-null mutants of frq, frh, wc1 and wc2 in Metarhizium robertsii, an insect pathogen. The authors described the subcellular localization of FRH and FRQ in wild-type and single-null backgrounds of each of the rest of the clock components, measure transcript expression and determine the interaction patterns among the four clock proteins, as well as phenotypic traits such as the formation of rings, the production of spores or the response to stress conditions.
There is a huge amount of work in this manuscript and will be, in my opinion, of interest for the readers of JoF. I only have some comments. The first one is the level of the language. There are long parts of the manuscript that need to be rewritten. Please, check the marked version of the manuscript I attach to the present report. It may help.
Second, maybe would be of interest for the authors to redraw the phylogenetic tree in Figure 1B to include more orthologs and to track the emergence of the Frq paralogs described in specific species.
Finally, a simple question: Do FRH and FRQ show the same nuclear localization before and after DAPI staining? If so, could the authors show, as supplementary material, nuclear localization before DAPI staining ?
Taking everything into consideration, my recommendation is Major Review

Author Response
Please see attached a file.

Round 2
Reviewer 2 Report
The authors have included most of the corrections I made to the first version of the manuscript. However, in that version I marked just some of the corrections to make. I still recommend the authors to review again the text and try to improve the quality of the language.